# Global Convergence of Gradient Descent for Asymmetric Low-Rank Matrix Factorization

**Tian Ye**
Institute for Interdisciplinary Information Sciences
Tsinghua University
yet17@mails.tsinghua.edu.cn

**Simon S. Du**
Paul G. Allen School of Computer Science and Engineering
University of Washington
ssdu@cs.washington.edu

## Abstract

We study the asymmetric low-rank factorization problem:

$$\min_{\mathbf{U}\in\mathbb{R}^{m\times d},\mathbf{V}\in\mathbb{R}^{n\times d}} \frac{1}{2}\|\mathbf{U}\mathbf{V}^\top - \mathbf{\Sigma}\|_F^2$$

where $\mathbf{\Sigma}$ is a given matrix of size $m \times n$ and rank $d$. This is a canonical problem that admits two difficulties in optimization: 1) non-convexity and 2) non-smoothness (due to unbalancedness of $\mathbf{U}$ and $\mathbf{V}$). This is also a prototype for more complex problems such as asymmetric matrix sensing and matrix completion. Despite being non-convex and non-smooth, it has been observed empirically that the randomly initialized gradient descent algorithm can solve this problem in polynomial time. Existing theories to explain this phenomenon all require artificial modifications of the algorithm, such as adding noise in each iteration and adding a balancing regularizer to balance the $\mathbf{U}$ and $\mathbf{V}$.

This paper presents the first proof that shows randomly initialized gradient descent converges to a global minimum of the asymmetric low-rank factorization problem with a polynomial rate. For the proof, we develop 1) a new symmetrization technique to capture the magnitudes of the symmetry and asymmetry, and 2) a quantitative perturbation analysis to approximate matrix derivatives. We believe both are useful for other related non-convex problems.

## 1   Introduction

This paper studies the asymmetric low-rank matrix factorization problem:

$$\min_{\mathbf{U}\in\mathbb{R}^{m\times d},\mathbf{V}\in\mathbb{R}^{n\times d}} f\left(\mathbf{U},\mathbf{V}\right) := \frac{1}{2}\|\mathbf{U}\mathbf{V}^\top - \mathbf{\Sigma}\|_F^2. \tag{1}$$

where $\mathbf{\Sigma}\in\mathbb{R}^{m\times n}$ is a given matrix of rank $d$. While solving this optimization problem is not hard (e.g., using power method), in this paper, we are interested in using randomly initialized gradient descent to solve this problem:

$$\mathbf{U}_{t+1} = \mathbf{U}_t + \eta(\mathbf{\Sigma} - \mathbf{U}_t\mathbf{V}_t^\top)\mathbf{V}_t; \tag{2}$$

$$\mathbf{V}_{t+1} = \mathbf{V}_t + \eta(\mathbf{\Sigma} - \mathbf{U}_t\mathbf{V}_t^\top)^\top\mathbf{U}_t, \tag{3}$$

35th Conference on Neural Information Processing Systems (NeurIPS 2021).

where $\eta > 0$ is the learning rate and $\mathbf{U}_0, \mathbf{V}_0$ are randomly initialized according to some distribution. Empirically, gradient descent with a *constant* learning rate can efficiently solve this problem (see, e.g., Figure 1 in Du et al. [2018]). Somehow surprisingly, there is no global convergence proof of this generic algorithm, let alone convergence rate analysis. The main difficulties are 1) the problem is non-convex and 2) this problem is not smooth with respect to $(\mathbf{U}, \mathbf{V})$ because the magnitudes of them can be highly unbalanced.

To motivate the study of gradient descent for this optimization problem, we note that this is a prototypical optimization problem that illustrates the gap between practice and theory. In particular, the prediction function $\mathbf{U}\mathbf{V}^\top$ is homogeneous: if we multiply a factor by a scalar $c$ and divide another factor by $c$, the prediction function remains the same. This homogeneity also exists in deep learning models. Therefore, progress made in understand (1) can further help us gain understanding on other non-convex problems, such as asymmetric matrix sensing, asymmetric matrix completion, and deep learning optimization. We refer readers to Du et al. [2018] for more discussions.

For Problem (1), Du et al. [2018] showed gradient flow (gradient descent with the step size $\eta \to 0$),

$$\dot{\mathbf{U}} = \left(\mathbf{\Sigma} - \mathbf{U}\mathbf{V}^\top\right)\mathbf{V} \text{ and } \dot{\mathbf{V}} = \left(\mathbf{\Sigma} - \mathbf{U}\mathbf{V}^\top\right)^\top \mathbf{U},$$

converges to the global minimum but no rate was given. Here $\mathbf{U} : \mathbb{R} \to \mathbb{R}^{m \times d}, \mathbf{U} = \mathbf{U}(t)$ and $\mathbf{V} : \mathbb{R} \to \mathbb{R}^{n \times d}, \mathbf{V} = \mathbf{V}(t)$ are an integral curve over manifold $\mathbb{R}^{m \times d} \times \mathbb{R}^{n \times d}$, and $\dot{\mathbf{U}} := \frac{d\mathbf{U}}{dt}$, $\dot{\mathbf{V}} := \frac{d\mathbf{V}}{dt}$. Key in their proof is an invariance maintained by gradient flow: $\frac{d}{dt}\left(\mathbf{U}^\top\mathbf{U} - \mathbf{V}^\top\mathbf{V}\right) = 0$. This invariance implies that if initially the difference between the magnitudes of $\mathbf{U}_0$ and $\mathbf{V}_0$ is small, then the difference remains small. This in turn guarantees the smoothness on the gradient flow trajectory. Du et al. [2018] further uses a geometric result (all saddle points in the objective function are strict and all local minima are global minima [Ge et al., 2015, 2017b, 2016, Li et al., 2019b]), and then invokes the stable manifold theorem to show the global convergence of gradient flow [Lee et al., 2016, Panageas and Piliouras, 2016].

However, to prove a polynomial convergence rate, the approach that solely relies on the geometry will fail because there exists a counter example [Du et al., 2017]. Furthermore, for gradient descent with $\eta > 0$, the key invariance no longer holds.[1]

Du et al. [2018] also studied gradient descent with decreasing step sizes $\eta_t = O\left(t^{-1/2}\right)$, and obtained an "approximate global optimality result": if the magnitude of the initialization is $O(\delta)$, then gradient descent converges to a $\delta$-optimal solution, i.e., this result does not establish that gradient descent *converges* to a global minimum. And again, there was no convergence rate. Furthermore, their result crucially relies on $\eta_t$ is of order $O\left(t^{-1/2}\right)$ to ensure the second order term does not diverge and thus does not apply to gradient descent with a *constant* learning rate.

Some previous works, e.g., Ge et al. [2015], Jin et al. [2017], modified the gradient descent algorithm to the *perturbed gradient descent algorithm* by adding an isotropic noise at each iteration, which can help escape strict saddle points and bypass the exponential lower bound in Du et al. [2017]. To deal with the non-smooth problem, they also added a balancing regularization term [Park et al., 2017, Tu et al., 2016, Ge et al., 2017a, Li et al., 2019b], $\frac{1}{8}\|\mathbf{U}^\top\mathbf{U} - \mathbf{V}^\top\mathbf{V}\|_F^2$ to the objective function to ensure balancedness between $\mathbf{U}$ and $\mathbf{V}$ throughout the optimization process. With these two modifications, one can prove a polynomial convergence rate. However, experiments suggest that the isotropic noise and the balancing regularizer may be proof artifacts, because vanilla gradient descent applies to the original objective function (1) without any regularizer finds a global minimum efficiently. From a practical point of view, one does not want to add noise or additional regularization because it may require more hyper-parameter tuning.

The only global quantitative analysis for randomly initialized gradient is by Du et al. [2018] who proved the global convergence rate for the case where $\mathbf{\Sigma}$ has rank $1$, and $\mathbf{U}$ and $\mathbf{V}$ are two vectors. In this case, one can reduce the problem to the dynamics of $4$ variables, which can be easily analyzed. Unfortunately, it is very difficult to generalize their analysis to the general rank setting. Recent work also studies this case with noise injected to the gradient update [Liu et al., 2021].

In this paper, we develop new techniques to overcome the technical difficulties and obtain the first polynomial convergence of randomly initialized gradient descent for solving the asymmetric low-rank

---

[1]While the invariance can still hold *approximately* in some way, characterizing the approximation error is highly non-trivial, and this is one of our key technical contributions.

matrix factorization problem. Most importantly, our analysis is completely different from existing ones: we give a thorough characterization of the entire trajectory of gradient descent.

Before presenting our main results, we emphasize that the goal of this paper is not to provide new provably efficient algorithms to solve Problem (1), but to provide a rigorous analysis of an intriguing and practically relevant phenomenon on gradient descent. This is of the same flavor as the recent breakthrough on understanding Burer-Moneiro method for solving semidefinite programs [Cifuentes and Moitra, 2019].

## 1.1 Main Results

Our main result is below.

**Theorem 1.1.** *Suppose each entry of $\mathbf{U}_0$ and $\mathbf{V}_0$ are initialized using Gaussian distribution with mean $0$ and variance $\varepsilon^2$, where $\varepsilon = \tilde{O}\left(\frac{\sigma_d}{\sqrt{d^3\sigma_1}(m+n)}\right)$.[2] Then there exists $T_{total}(\delta, \eta) = O\left(\frac{1}{\eta\sigma_d}\ln\frac{d\sigma_d}{\varepsilon} + \frac{1}{\eta\sigma_d}\ln\frac{\sigma_d}{\delta}\right)$ such that for any $\delta > 0$ and learning rate $\eta = O\left(\frac{\sigma_d\varepsilon^2}{d\sigma_1^3}\right)$, we have that with high probability over the initialization, when $t > T_{total}(\delta, \eta), f(\mathbf{U}_t, \mathbf{V}_t) \leq \delta$.*

Here, $\sigma_1$ and $\sigma_d$ are the largest and the smallest singular values of $\mathbf{\Sigma}$, respectively. Notably, in sharp contrast to the result in Du et al. [2018], which requires the initialization depends on $\delta$, our initialization does not depend on the target accuracy. To our knowledge, this is the first global convergence result for gradient descent in solving Problem (1). Furthermore, we give a polynomial rate. The first term in $T_{\text{total}}(\delta, \eta)$ represents a warm-up phase and the second term represents the local linear convergence phase, which will be clear in the analysis sections. On the other hand, while we believe $T_{\text{total}}(\delta, \eta)$ is nearly tight, our requirement for $\eta$ is loose. An interesting future direction is further relax this requirement.

Now by taking $\eta \to 0$, we have the following corollary for gradient flow.

**Corollary 1.2.** *Given $\delta > 0$, there exists $T = O\left(\frac{1}{\sigma_d}\ln\frac{d\sigma_d}{\varepsilon} + \frac{1}{\sigma_d}\ln\frac{\sigma_d}{\delta}\right)$, such that with high probability over the initialization, for all $t \geq T$, we have $f(\mathbf{U}_t, \mathbf{V}_t) \leq \delta$.[3]*

This is also the first convergence rate result of randomly initialized gradient flow for asymmetric matrix factorization. We note that our analysis on gradient flow is nearly tight. To see this, consider the ordinary differential equation $\dot{a}_t = (\sigma_d - a_t^2)a_T$ with initial point $a_0 > 0$, then $s = a^2$ has analytical solution $s_t = \frac{\sigma_d e^{2\sigma_d t}}{e^{2\sigma_d t} + \frac{\sigma_d}{a_0^2} - 1}$. Hence, to achieve a $\delta$ optimal solution, i.e. $|\sigma_d - a^2| \leq \delta$, we need $\sigma_d\left(\frac{\sigma_d}{a_0^2} - 1\right)\frac{1}{\delta} \leq e^{2\sigma_d t} + \frac{\sigma_d}{a_0^2} - 1$. Hence $T = \Theta\left(\frac{1}{\sigma_d}\ln\frac{\sigma_d}{\delta}\right)$ is necessary.

**Remark 1.2.** *We note that the weights will also converge. In section 3.3, we will prove that $\mathbf{U}$ and $\mathbf{V}$ are always bounded, which implies the gradient norm is bounded in terms of the loss. Since the loss function converges linearly, we have that the gradient norm converges to $0$ linearly. The convergence of $U$ and $V$ follows from that the trajectory of $(\mathbf{U}_t, \mathbf{V}_t)$ forms a Cauchy sequence.*

## 1.3 Additional Related Work

Here we discuss additional related work. First, in the symmetric setting, e.g., $\min_{\mathbf{U}} \|\mathbf{U}\mathbf{U}^\top - \mathbf{\Sigma}\|_F^2$, global convergence of randomly initialized gradient has been established in various settings [Jain et al., 2017, Li et al., 2018, Chen et al., 2019].[4] However, as has been highlighted in Li et al. [2019a,b], Park et al. [2017], Tu et al. [2016], generalization to the asymmetric case is highly non-trivial. The major technical difficulty is to deal with the unbalancedness between $\mathbf{U}$ and $\mathbf{V}$. To prevent this, additional balancing regularization is often added [Li et al., 2019b, Park et al., 2017, Tu et al., 2016, Sun and Luo, 2016], though empirically this has been shown to be unnecessary.

Another line of work showed one can first uses spectral initialization to find a near-optimal solution, then starting from there, gradient descent converges to an optimum with a linear rate [Tu et al.,

---

[2]$\tilde{O}$ hides logarithmic terms.

[3]In gradient flow, $t$ is a continuous time index.

[4]In Appendix B, we show the dynamics of gradient flow actually admits a *closed form*, and thus can be easily analyzed.

2016, Zheng and Lafferty, 2016, Zhao et al., 2015, Bhojanapalli et al., 2016], though in practice random initialization often suffices. Recently, Ma et al. [2021] proved that if 1) the initialization is close to a global minimum and 2) $\mathbf{U}$ and $\mathbf{V}$ are balanced, then without adding additional balancing regularizer, gradient descent converges to a global minimum. Our stage two's analysis is similar to theirs. However, their result cannot be directly applied to our analysis because they require a more stringent initialization than our stage two's initial point. There is also other work studying the balancing effect in terms of the learning rate [Wang et al., 2021].

**Notations.** Throughout the paper, bold letters, e.g., $\mathbf{U}, \mathbf{V}, \boldsymbol{\Sigma}$, are reserved for matrices running in the algorithm, non-bold letters, e.g., $U, V, \Sigma$ are for our analysis. For a matrix $W$ with rank $r$, denote $\sigma_i(W)$ as the $i^{\text{th}}$ largest singular value of $W$, $\forall i \in [r]$. Furthermore, if $W$ is symmetric, denote $\lambda_i(W)$ as the $i^{\text{th}}$ largest eigenvalue of $W$. Let $\boldsymbol{\Sigma} \in \mathbb{R}^{m \times n}$ be a rank-$d$ matrix with singular value $\sigma_1 \geq \cdots \geq \sigma_d > 0$, and define its conditional as $\kappa := \frac{\sigma_1}{\sigma_d}$. Our goal is to factorize $\boldsymbol{\Sigma}$ into $\mathbf{U}\mathbf{V}^\top$.

## 2 Main Difficulties and Technique Overview

### 2.1 A Reduction to Principal and Complement Spaces.

The starting point is the Polyak-Łojasiewicz condition: if we can establish that $\max\{\sigma_d(\mathbf{U}_t), \sigma_d(\mathbf{V}_t)\}$ is lower bounded by a considerable constant $c_{\max}$, then we have $\|\nabla f(\mathbf{U}, \mathbf{V})\| \geq c_{\max}\sqrt{2f(\mathbf{U}, \mathbf{V})}$, which implies a linear convergence. However, the $d^{\text{th}}$ singular values of $\mathbf{U}$ and $\mathbf{V}$ are not monotonic with $t$, and they can even decrease to an extremely small value.

To deal with this issue, we consider the following transformation. Let the singular value decomposition of $\boldsymbol{\Sigma}$ is $\boldsymbol{\Sigma} \equiv \Phi\boldsymbol{\Sigma}'\Psi^\top$, where $\Phi \in \mathbb{R}^{m \times m}$ and $\Psi \in \mathbb{R}^{n \times n}$ are unitary matrices, and $\boldsymbol{\Sigma}'$ is diagonal matrix. Define $\mathbf{U}'_t := \Phi^{-1}\mathbf{U}_t$ and $\mathbf{V}'_t = \Psi^{-1}\mathbf{V}_t$. Then we can rewrite equations (2) and (3) as

$$\mathbf{U}'_{t+1} = \mathbf{U}'_t + \eta(\boldsymbol{\Sigma}' - \mathbf{U}'_t\mathbf{V}'^\top_t)\mathbf{V}'_t; \tag{4}$$

$$\mathbf{V}'_{t+1} = \mathbf{V}'_t + \eta(\boldsymbol{\Sigma}' - \mathbf{U}'_t\mathbf{V}'^\top_t)^\top\mathbf{U}'_t. \tag{5}$$

Hence, without loss of generality, we can assume $\boldsymbol{\Sigma}$ is a diagonal matrix with $\boldsymbol{\Sigma}_{i,i} = \sigma_i$, $\forall i \in [d]$, and $\boldsymbol{\Sigma}_{i,j} = 0$ otherwise.

To proceed, we will analyse the principal space and the complement space separately. We denote the upper $d \times d$ matrix of $\mathbf{U}$ as $U$ and denote the lower $(m-d) \times d$ matrix of $\mathbf{U}$ as $J$. Similarly, we define the upper $d \times d$ matrix of $\mathbf{V}$ as $V$ and the lower $(n-d) \times d$ matrix as $K$. Define $\Sigma := \text{diag}(\sigma_1, \cdots, \sigma_d)$. We can write out the dynamics of these matrices:

$$U_{t+1} = U_t + \eta(\Sigma - U_tV_t^\top)V_t - \eta U_tK_t^\top K_t; \tag{6}$$

$$V_{t+1} = V_t + \eta(\Sigma - U_tV_t^\top)^\top U_t - \eta V_tJ_t^\top J_t; \tag{7}$$

$$J_{t+1} = J_t - \eta J_t(V_t^\top V_t + K_t^\top K_t); \tag{8}$$

$$K_{t+1} = K_t - \eta K_t(U_t^\top U_t + J_t^\top J_t). \tag{9}$$

**Additional Notations** Throughout this paper, we have some notation conventions. First of all, if we omit the subscript (iteration number) of a matrix, then it represents that this matrix at any iteration $t$. If some matrices without subscripts appear in the same equation, it means the equation holds for arbitrary iteration $t$, and the subscript for each matrix should be the same. For instance, if we define $A := \frac{U+V}{2}$, it means we define $A_t := \frac{U_t+V_t}{2}$, $\forall t \geq 0$. Similarly, we define $B := \frac{U-V}{2}$.

Besides $A, B, J$ and $K$, there are some other special capital letters used to represent specific matrices throughout this paper. Here is a list.

$$S = AA^\top;$$
$$P = \Sigma - AA^\top + BB^\top;$$
$$Q = AB^\top - BA^\top.$$

We define such $S$ is because in symmetric case ($B \equiv 0$), although it is hard to find analytical solution for $A$ in continuous time case, we do find analytical form for $S$, which contains all information about the singular values of $A$.

$P$ and $Q$ are just the symmetric and skew-symmetric part of matrix $\Sigma - UV^\top$. Hence the linear convergence of gradient descent is equivalent the linearly diminishing of $P$ and $Q$ by Pythagorean theorem. We will mention their definitions every time we use them.

## 2.2 Symmetrization

Our key observation is that although the singular values of $U$ and $V$ may not have monotonic property, the *symmetrized matrix* has this property. Formally, we define

$$A := \frac{U+V}{2} \quad \text{and} \quad B := \frac{U-V}{2}.$$

Here, $A$ represents the magnitude in the principal space and $B$ represents the magnitude of asymmetry. Empirically, we can observe that by choosing a sufficiently small learning rate $\eta$, we have two desired properties:

1. The smallest singular value of $A$ is almost monotonically increasing;

2. The norms of $B, J, K$ are almost monotonically decreasing.

The first property ensures we are learning the "signal", $\Sigma$, and the second property ensures the "noise" is disappearing. Therefore, if we can establish these two properties, we can prove the global convergence.

## 2.3 Two Stage Analysis

The analysis for asymmetric low rank case is divided into two stages. In the first stage we mainly focus on the increasing rate of $\sigma_d(A)$. We will prove that in gradient descent method $\sigma_d(A_d)$ increases exponentially fast to $\sqrt{\frac{\sigma}{2}}$ and then $\|P\|_{op}$ drops exponentially fast to $\frac{\sigma_d}{4}$, while preserving $\|B\|_F, \|J\|_{op}$ and $\|K\|_{op}$ small. In the second stage, we will use the large $\sigma_d(A)$ to lower bound the convergence speed of $\|\mathbf{\Sigma} - \mathbf{U}\mathbf{V}^\top\|_F^2$. We will prove that, once gradient descent starts at a point with small $\|P\|_{op}, \|B\|_F, \|J\|_{op}$ and $\|K\|_{op}$, it will converge to global optimal point exponentially fast.

# 3 Proof Sketch of Theorem 1.1

## 3.1 Initialization

We first use a Gaussian distribution to generate matrices $U, V, J, K$ element-wisely and independently[5]. By standard random matrix theory (Corollary 2.3.5 and Theorem 2.7.5 of Tao [2012]), we know that $\exists c > 0$, such that with high probability, the smallest singular value of $\frac{U+V}{2}$ is larger than $\frac{1}{c\sqrt{d}}$, the largest singular value of $\frac{U+V}{2}$ is smaller than $c\sqrt{d}$, the Frobenius norm of $B$ is less than $cd$ and the operator norms of $J$ and $K$ are less than $c\sqrt{\max\{m', d\}}$ and $c\sqrt{\max\{n', d\}}$, respectively, where $m' = m - d, n' = n - d$.

The initializations $U_0, V_0, J_0, K_0$ are then scaled by $\varepsilon$ where $\varepsilon$ specified in Theorem 1.1.

## 3.2 Stage One: Warm-Up Phase

In this stage, we would like to prove the following theorem.

**Theorem 3.1.** *By choosing $\varepsilon = \tilde{O}\left(\frac{\sigma_d}{\sqrt{d^3 \sigma_1}(m+n)}\right)$ and $\eta = O\left(\frac{\sigma_d \varepsilon^2}{d\sigma_1^3}\right)$, we have that there exists $T_0 = O\left(\frac{1}{\eta \sigma_d} \ln \frac{d\sigma_d}{\varepsilon^2}\right)$, such that $\forall t \leq T_0$,*

- $\frac{\varepsilon^2}{c^2 d} I \preceq A_t A_t^\top \preceq 2\Sigma$;

- $\|B_t\|_F \leq 2cd\varepsilon$;

---

[5]Strictly speaking, we cannot make any assumption on $U, V, J, K$ since they need information of singular value decomposition of $\mathbf{\Sigma}$. However, a random generation of $\mathbf{U}$ and $\mathbf{V}$ implies a random generation of $U, V, J, K$ because we use unitary transformations.

- $\sigma_d(A_{T_0}) \geq \sqrt{\frac{\sigma_d}{2}}$;

- $\sigma_1(P_{T_0}) \leq \frac{\sigma_d}{4}$;

- $\|J_t\|_{op} \leq c\varepsilon\sqrt{\max\{m', d\}}$, $\|K_t\|_{op} \leq c\varepsilon\sqrt{\max\{n', d\}}$.

We first give some intuitions about the five conditions in Theorem 3.1. The first condition represents the "signal" is properly bounded from below and above throughout stage one. The second condition shows the magnitude of asymmetry is small throughout stage one. We note that it is crucial to study the Frobenius norm of $B$ instead of operator norm, because Frobenius norm admits a nice expansion for analysis. The third condition is an important one, which guarantees after $T_0$ iterations, we have enough "signal" strength in the principal space. The fourth condition is a technical one, which represents the symmetric error is small after $T_0$ iterations. The fifth condition represents the magnitude of the complement space remains small.

**Proof Sketch.** The proof of Theorem 3.1 is quite challenging and require new technical ideas and careful calculations, which we explain below.

To analyze the dynamic of $\sigma_d(A)$, let us recall how it behaves in the continuous-time case. Our main idea is that, instead of analyzing $A$ itself, we consider the symmetric matrix $S := AA^\top$. Then $\frac{dS}{dt} \approx (\Sigma - S)S + S(\Sigma - S)$ plus some small perturbation terms about $B$, $J$ and $K$.

If we only consider a differential equation $\dot{S} = (\Sigma - S)S + S(\Sigma - S)$, a well-known theorem (Theorem 12 in Lax [2007]) shows that if the singular values of $S$ are different from each other, and $\xi$ is the singular vector that $S\xi = \sigma_d(S)\xi$, then the derivative of $\sigma_d(S)$ is exactly $\xi^\top \dot{S}\xi$, which is lower bounded by $2(\sigma_d - \sigma_d(S))\sigma_d(S)$. To adapt it to discrete case, we prove the following lemma.

**Lemma 3.2.** *Suppose $S, \Sigma \in \mathbb{R}^{d \times d}$ are two definite positive matrices, $\eta > 0$, and $S' = (I + \eta(\Sigma - S))S(I + \eta(\Sigma - S))$. Suppose $\sigma_1(S) \leq 2\sigma_1, \sigma_d(\Sigma) \geq \sigma_d$ and $\sigma_1(\Sigma) \leq \sigma_1$. Define $s = \sigma_d(S)$ and $s' = \sigma_d(S')$. Then $\forall \beta \in (0, 1)$ and $\eta \leq \frac{\beta}{8\sigma_1}$,*

$$s' \geq (1 + \eta(\sigma_d - s))^2 s - \frac{8 + 6\beta}{1 - \beta}\sigma_1^3 \eta^2.$$

This lemma shows if we ignore perturbations from $B$, $J$ and $K$, then for small $\eta$ (when $\eta^2$ is of smaller order than the first term), the least eigenvalue of $S$ increases at a *geometric* rate.

However, there are also some small perturbation terms about $B$, $J$ and $K$ while doing analysis. $\|J\|_{op}$ and $\|K\|_{op}$ are easy to give an upper bound, since by (8) and (9), we know that by choosing small enough $\eta$, they are monotonically decreasing. However, the dynamic of $B$ is highly non-trivial. After some careful calculations (cf. (19)), we find that the increasing rate of $\|B\|_F^2$ is related to the smallest eigenvalue of $P := \Sigma - AA^\top + BB^\top$: if $\max\{0, -\lambda_d(P)\}$ is small, then $\|B\|_F^2$ increases slowly.

Now we would like to give a lower bound on $\lambda_d(P)$. Inspired by gradient flow case, $P$ and $S$ are almost complementary of each other, and their dynamic behaves similarly. Hence we have $\dot{P} \approx -(\Sigma - P)P - P(\Sigma - P)$ with some small perturbation terms about $B$, $J$ and $K$. Hence we can use lemma 3.3 to give a lower bound in discrete case.

**Lemma 3.3.** *Suppose $P, \Sigma \in \mathbb{R}^{d \times d}$ are two symmetric matrices, $\eta > 0$, and $P' = (I - \eta(\Sigma - P))P(I - \eta(\Sigma - P))$. Suppose $\sigma_1(P) \leq 2\sigma_1$ and $\sigma_d I \preceq \Sigma \preceq \sigma_1 I$. Define $p = \lambda_d(P)$ and $p' = \lambda_d(P')$. Then $\forall \beta \in (0, 1)$ and $\eta \leq \frac{\beta}{8\sigma_1}$,*

$$p' \geq \begin{cases} (1 - \eta\sigma_d)^2 p - \frac{8+6\beta}{1-\beta}\sigma_1^3\eta^2, & \text{if } p < 0; \\ 0, & \text{if } p \geq 0. \end{cases}$$

Notice that we use $B$ while analyzing $P$ and use $P$ while analyzing $B$. Hence, during the whole process, we need to bound both of them inductively.

Finally, once $\sigma_d(A)$ increases to a relatively large amount, we can use it to prove that $\|P\|_{op}$ will decrease exponentially fast to $\frac{\sigma_d}{4}$. One cannot simply prove that $P$ converges to zero in this stage, since the perturbation term $B$ will never converge to zero.

Below we give more details.

### 3.2.1 Assumptions

We make some assumptions on $A$ and $B$ in iterations $t \leq T_0$, where $T_0$ will be defined at the end of subsubsection 3.2.4, and we will verify the assumptions in the end.

(1) $\frac{\varepsilon^2}{c^2 d} I \preceq A A^\top \preceq 2\Sigma$.

(2) The Frobenius norm of $B$ is bounded by $e_b d\varepsilon$ for some $e_b \geq c$, where $e_b$ will be determined later[6]. Hence its operator norm is also bounded by $e_b d\varepsilon$.

### 3.2.2 Dynamics on $A$, $B$ and $P$

The dynamics on $J$ and $K$ is trivial, since by equations (8) and (9), i.e.

$$J_{t+1} = J_t - \eta J_t(V_t^\top V_t + K_t^\top K_t);$$
$$K_{t+1} = K_t - \eta K_t(U_t^\top U_t + J_t^\top J_t),$$

we know that if we choose $\eta \leq \frac{1}{3\sigma_1}$, one can inductively proved that $0 \preceq V_t^\top V_t + K_t^\top K_t \preceq 3\sigma_1 I$ and $0 \preceq U_t^\top U_t + J_t^\top J_t \preceq 3\sigma_1 I$ by using the first two assumptions in subsection 3.2.1. And then it follows that the operator norms of $J$ and $K$ are monotonically decreasing in this stage.

However, it is non-trivial to prove that $\|B\|_{op}$ keeps small. We will analyze the dynamics of $A, B$ and $P := \Sigma - AA^\top + BB^\top$ together inductively.

First of all, from equations (6) and (7), we can write down the dynamics of $A := \frac{U+V}{2}$ and $B := \frac{U-V}{2}$ as following.

$$
\begin{aligned}
A_{t+1} &= A_t + \eta(\Sigma - A_t A_t^\top + B_t B_t^\top)A_t - \eta(A_t B_t^\top - B_t A_t^\top)B_t \\
&\quad -\eta A_t \frac{K_t^\top K_t + J_t^\top J_t}{2} - \eta B_t \frac{K_t^\top K_t - J_t^\top J_t}{2};
\end{aligned}
\tag{10}
$$

$$
\begin{aligned}
B_{t+1} &= B_t - \eta(\Sigma - A_t A_t^\top + B_t B_t^\top)B_t + \eta(A_t B_t^\top - B_t A_t^\top)A_t \\
&\quad -\eta A_t \frac{K_t^\top K_t - J_t^\top J_t}{2} - \eta B_t \frac{K_t^\top K_t + J_t^\top J_t}{2}.
\end{aligned}
\tag{11}
$$

We can further calculate

$$
\begin{aligned}
P_{t+1} &= P_t - \eta P_t(\Sigma - P_t) - \eta(\Sigma - P_t)P_t + \eta^2(P_t P_t P_t - P_t \Sigma P_t) - 2\eta B_t B_t^\top P_t \\
&\quad -2\eta P_t B_t B_t^\top - \eta(A_t + \eta P_t A_t)C_t^\top - \eta C_t(A_t + \eta P_t A_t)^\top - \eta^2 C_t C_t^\top \\
&\quad +\eta(B_t + \eta P_t B_t)D_t^\top + \eta D_t(B_t + \eta P_t B_t)^\top + \eta^2 D_t D_t^\top
\end{aligned}
\tag{12}
$$

where

$$
C_t := -A_t B_t^\top B_t + B_t A_t^\top B_t - A_t \frac{K_t^\top K_t + J_t^\top J_t}{2} - B_t \frac{K_t^\top K_t - J_t^\top J_t}{2};
\tag{13}
$$

$$
D_t := +A_t B_t^\top A_t - B_t A_t^\top A_t - A_t \frac{K_t^\top K_t - J_t^\top J_t}{2} - B_t \frac{K_t^\top K_t + J_t^\top J_t}{2},
\tag{14}
$$

are two small perturbation terms.

### 3.2.3 Dynamics on $A$

Given (10), we can give a lower bound for the minimal singular value of $A_{t+1}$.

$$
\begin{aligned}
\sigma_d(A_{t+1}) &\geq \sigma_d(A_t + \eta(\Sigma - A_t A_t^\top)A_t) \\
&\quad -\eta \left\| B_t B_t^\top A_t - A_t B_t^\top B_t + B_t A_t^\top B_t - A_t \frac{K_t^\top K_t + J_t^\top J_t}{2} - B_t \frac{K_t^\top K_t - J_t^\top J_t}{2} \right\|_{op}.
\end{aligned}
$$

---

[6] We will show later that it is appropriate to choose $e_b = 2c$.

For the first part, we could define $S_t := A_t A_t^\top$, and $\overline{S}_{t+1} := (I + \eta(\Sigma - S_t))S_t(I + \eta(\Sigma - S_t))$. Then according to lemma 3.2, by choosing $\beta = \frac{1}{2}$ and $\eta \le \frac{1}{16\sigma_1}$, we have[7]

$$
\begin{aligned}
\sigma_d(A_{t+1}) \ge\ & \sqrt{(1 + \eta(\sigma_d - \sigma_d(A_t)^2))^2 \sigma_d(A_t)^2 - 22\sigma_1^3 \eta^2} \\
& -1.5\sqrt{2\sigma_1}\eta(e_b^2 + c^2)\varepsilon^2(m+n)d.
\end{aligned} \tag{15}
$$

For simplicity, we denote $\sigma_d(A_t)$ by $a_t$, and define $s_t = \sigma_d(S_t) = a_t^2$.

After some routine computations[8], we can prove that it takes at most $T_1 := O\left(\frac{1}{\eta\sigma_d}\ln\frac{d\sigma_d}{\varepsilon^2}\right)$ iterations to make $a_t$ to at least $\sqrt{\frac{\sigma_d}{2}}$, and additional computations show that, if $a_t$ is always bounded by $\sqrt{2\sigma_1}$, then once $a_t$ becomes larger than $\sqrt{\frac{\sigma_d}{2}}$, it is always larger than $\sqrt{\frac{\sigma_d}{2}}$.

### 3.2.4 Dynamics on $P$

To bound $P_t$ by equation (12), we need to first bound the norms of $C_t$ and $D_t$ by (13) and (14). By simple triangle inequalities we have[9]

$$
\begin{aligned}
\|C_t\|_{op} &\le\ \sqrt{2\sigma_1}(e_b^2 + c^2)(m+n)d\varepsilon^2; \\
\|D_t\|_{op} &\le\ 8\sigma_1 e_b d\varepsilon,
\end{aligned}
$$

where the last inequality holds when choosing $\varepsilon \le \frac{\sqrt{\sigma_1}e_b d}{c^2(m+n)}$. Then we can conclude that

$$
P_{t+1} = (I - \eta(\Sigma - P_t))P_t(I - \eta(\Sigma - P_t)) + E_t, \tag{16}
$$

where $E_t$ is a matrix with operator norm less than $O(\eta^2\sigma_1^3 + \eta e_b^2\varepsilon^2(m+n)d\sigma_1 + \eta^2\sigma_1^2 e_b^2 d^2\varepsilon^2)$. By choosing $\varepsilon \le \frac{\sqrt{\sigma_1}}{e_b d}$ and $\eta \le \frac{c^2\varepsilon^2(m+n)d}{\sigma_1^2}$, we have $\|E_t\|_{op} \le O(\eta e_b^2\varepsilon^2(m+n)d\sigma_1)$. Further more, by choosing $\beta = \frac{1}{2}$ and $\eta \le \frac{1}{16\sigma_1}$ in lemma 3.3, we have

$$
\lambda_d(P_{t+1}) \ge \max\left\{(1 - \eta\sigma_d)^2\lambda_d(P_t) - O(\eta e_b^2\varepsilon^2(m+n)d\sigma_1), -O(\eta e_b^2\varepsilon^2(m+n)d\sigma_1)\right\}.
$$

Because $P_0$ is initially positive, we know that

$$
\lambda_d(P_t) \ge -O(e_b^2\varepsilon^2(m+n)d\kappa). \tag{17}
$$

This lower bound verifies the assumption that $A_t A_t^\top \preceq 2\Sigma$, since $A_t A_t^\top = \Sigma - P + B_t B_t^\top \preceq \Sigma + O(e_b^2\varepsilon^2(m+n)d\kappa)I + e_b^2\varepsilon^2 dI \preceq 2\Sigma$ by choosing $e_b^2\varepsilon^2 = O\left(\frac{\sigma_d}{(m+n)d\kappa}\right)$.

On the other hand, we can also analyze the operator norm of $P$ by using formula (16), since $\sigma_d(A_t) \ge \sqrt{\frac{\sigma_d}{2}}$ for $t \ge T_1$. This implies that

$$
\sigma_1(P_{t+1}) \le \left(1 - \frac{\eta\sigma_d}{2}\right)^2\sigma_1(P_t) + O(\eta e_b^2\varepsilon^2(m+n)d\sigma_1),
$$

and it follows that

$$
\sigma_1(P_{t+T_1}) \le \left(1 - \frac{\eta\sigma_d}{2}\right)^{2t}\sigma_1(P_{T_1}) + O(e_b^2\varepsilon^2(m+n)d\kappa). \tag{18}
$$

Inequality (18) shows that we only need at most $T_2 := O\left(\frac{1}{\eta\sigma_d}\ln\kappa\right)$ iterations after $T_1$ to make $\sigma_1(P_t) \le \frac{\sigma_d}{4}$. Because $\varepsilon^2 \le \sigma_d$, we have the total number of iteration $T_0 := T_1 + T_2 = O\left(\frac{1}{\eta\sigma_d}\ln\frac{d\sigma_d}{\varepsilon^2}\right)$.

---

[7]Please see (20) for the full steps for this inequality.

[8]Please see section D for details.

[9]Please see (21) for full steps for this inequality.

### 3.2.5 Dynamics on *B*

To verify the assumption about $\|B\|_F$ made in subsection 3.2.1, we cannot simply use the equation (11), since the error term $\|(AB^\top - BA^\top)A\|_{op}$ is approximately $O(\sigma_1\|B\|_{op})$, which will perturb the analysis seriously. Inspired by the continuous case that $\|\dot{B}\|_F^2 = 2\left\langle B, \dot{B}\right\rangle = \text{Tr}(B^\top PB) - \frac{1}{2}\|Q\|_F^2 \leq \text{Tr}(B^\top PB)$, where $Q = AB^\top - BA^\top$ if we assume $J = K = 0$. In this inequality, we hide the term $(AB^\top - BA^\top)AB^\top$ in $-\frac{1}{2}\|Q\|_F^2$ and wipe it completely in our analysis.

Hence, for discrete case, we have the following inequality[10],

$$
\begin{aligned}
\|B_{t+1}\|_F^2 - \|B_t\|_F^2 \leq\ & -2\eta\lambda_d(P_t)\|B_t\|_F^2 + \eta\|B_t^\top A_t\|_F\|K_t^\top K_t - J_t^\top J_t\|_F \\
& +\eta^2\|(\Sigma - A_t A_t^\top + B_t B_t^\top)B_t + (A_t B_t^\top - B_t A_t^\top)A_t \\
& -A_t\frac{K_t^\top K_t - J_t^\top J_t}{2} - B_t\frac{K_t^\top K_t + J_t^\top J_t}{2}\|_F^2 \\
\leq\ & O(\eta e_b^2\varepsilon^2(m+n)d\kappa)\|B_t\|_F^2 + O(\eta\sqrt{\sigma_1}e_b(m+n)d^2\varepsilon^3),
\end{aligned}
\tag{19}
$$

where the last equation is because we have chosen $\eta = O\left(\frac{\sigma_d\varepsilon^2}{d\sigma_1^3}\right)$ and $e_b^2\varepsilon^2 = O\left(\frac{\sigma_d}{(m+n)d\kappa}\right)$.

By some routine calculations in section E, we have that by choosing $\varepsilon = \tilde{O}\left(\frac{\sigma_d}{\sqrt{\sigma_1}e_b(m+n)}\right)$, it is appropriate to choose $e_b = 2c$, so that $\|B_T\|_F^2 \leq e_b^2 d^2\varepsilon^2$, and induction holds.

### 3.3 Stage Two: Local Convergence Phase

We have proved in theorem 3.1 that the gradient descent achieved a pretty good point at $T_0$, i.e. $\|B_{T_0}\|_F \leq 2cd\varepsilon$ and $\sigma_1(P_{T_0}) \leq \frac{\sigma_d}{4}$. In this subsection, we will prove that start from this point, the gradient descent will converge linearly to the global optimal point. Then theorem 1.1 follows.

We will prove inductively on the following conditions:

(1) $\|B\|_F = O(\frac{\sigma_d}{\sqrt{\sigma_1}})$;

(2) $\Delta_t := \|\Sigma - U_{T_0+t}V_{T_0+t}^\top\|_{op} \leq \left(1 - \frac{\eta\sigma_d}{2}\right)^t \frac{2}{5}\sigma_d$;

(3) $\sigma_d(U), \sigma_d(V) \geq \sqrt{\frac{\sigma_d}{2}}$.

Intuitively, the (1) guarantees the magnitude of asymmetry remains small; (2) guarantees that in the principal space, the error converges to 0 with a geometric rate; and (3) guarantees the "signal" in the principal space remains lower bounded.

**Proof Sketch.** First of all, it is easy to prove linear convergence of $J$ and $K$ by using assumption (3). Now we can verify the assumptions inductively.

(1) + (2) $\Rightarrow$ (3): We can prove $\sigma_d(UV^\top) = \Theta(\sigma_d)$. Because $U - V$ is small, (3) follows by triangle inequality.

(1) + (3) $\Rightarrow$ (2): Consider continuous-time case, if we assume $J = K = 0$, the time derivative of $\Sigma - UV^\top$ is $-(\Sigma - UV^\top)VV^\top - UU^\top(\Sigma - UV^\top)$. Hence the convergence rate is lower bounded by $\sigma_d(U)$ and $\sigma_d(V)$. Because the perturbation term $J$ and $K$ decreases exponentially, assumption (2) follows naturally. We transform this intuition to the discrete-time case.

(2) + (3) $\Rightarrow$ (1): Again, we use (19) to show that the increasing rate of $\|B\|_F^2$ is bounded by $\Delta$. Because $\Delta$ decreases exponentially, $\|B\|_F^2$ cannot diverge to infinity, but increase by a $\text{poly}(m, n, \kappa)$ factor. Then by taking $\varepsilon$ sufficiently small can we verify the assumption (1).

The full proof is deferred to appendix.

---

[10]Please see (22) for full steps for this inequality.

**Summary of Stage 2.** To sum up, we have $\|\mathbf{\Sigma} - \mathbf{U}_t \mathbf{V}_t^\top\|_F^2 = \|\Sigma - U_t V_t^\top\|_F^2 + \|U_t K_t^\top\|_F^2 + \|J_t V_t^\top\|_F^2 + \|J_t K_t^\top\|_F^2$, which can be further bounded by

$$
\begin{aligned}
\|\mathbf{\Sigma} - \mathbf{U}_{T_0+t} \mathbf{V}_{T_0+t}^\top\|_F^2 &\leq \left(1 - \frac{\eta\sigma_d}{2}\right)^t \frac{2}{5}\sigma_d + 2(c^2 + c^4)\varepsilon^2 \sigma_1(m+n)d\left(1 - \frac{\eta\sigma_d}{2}\right)^{2t} \\
&\leq C\left(1 - \frac{\eta\sigma_d}{2}\right)^t \sigma_d,
\end{aligned}
$$

for some universal constant $C$. Hence one only needs $T_f := O\left(\frac{\ln\frac{\sigma_d}{\delta}}{\eta\sigma_d}\right)$ iterations after $T_0$ to achieve an $\delta$-optimal point.

## 4  Conclusion

This paper proved that randomly initialized gradient descent converges to a global minimum of the asymmetric low-rank matrix factorization problem with a polynomial convergence rate. This result explains the empirical phenomena observed in prior work, and confirms that gradient descent with a constant learning rate still enjoys the auto-balancing property as argued in Du et al. [2018].

We believe our requirement of the step size $\eta$ is loose and a tighter analysis may improve the running time of gradient descent. Another interesting direction is to apply our techniques to other related problems such as asymmetric matrix sensing, asymmetric matrix completion and linear neural networks.[11]

## Acknowledgement

Simon S. Du gratefully acknowledges funding from NSF Award's IIS-2110170 and DMS-2134106.

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
