# A    Omitted Derivations of Formulas

We have omitted a number of complicated formulas in the main text to provide clear intuition and concise proof sketch. We will list all mentioned formulas here for readers' reference.

$$
\begin{aligned}
\sigma_d(A_{t+1}) &\geq \sigma_d(A_t + \eta(\Sigma - A_t A_t^\top)A_t) - \eta\left(3\sqrt{2\sigma_1}\|B_t\|_{op}^2 + \sqrt{2\sigma_1}(\|K_t\|_{op}^2 + \|J_t\|_{op}^2)\right) \\
&\geq \sqrt{\sigma_d(\overline{S}_{t+1})} - \eta\left(3\sqrt{2\sigma_1}e_b^2\varepsilon^2 d^2 + \sqrt{2\sigma_1}c^2\varepsilon^2(m+n)\right) \\
&\geq \sqrt{(1 + \eta(\sigma_d - \sigma_d(A_t)^2))^2 \sigma_d(A_t)^2 - 22\sigma_1^3\eta^2} \\
&\quad -1.5\sqrt{2\sigma_1}\eta(e_b^2 + c^2)\varepsilon^2(m+n)d.
\end{aligned}
\tag{20}
$$

$$
\begin{aligned}
\|C_t\|_{op} &\leq 2\sqrt{2\sigma_1}e_b^2 d^2\varepsilon^2 + \sqrt{2\sigma_1}c^2\varepsilon^2\left(\max\{d,m'\} + \max\{d,n'\}\right) \\
&\leq \sqrt{2\sigma_1}(e_b^2 + c^2)(m+n)d\varepsilon^2; \\
\|D_t\|_{op} &\leq 4\sigma_1 e_b d\varepsilon + \sqrt{2\sigma_1}c^2\varepsilon^2\left(\max\{d,m'\} + \max\{d,n'\}\right) \\
&\leq 8\sigma_1 e_b d\varepsilon.
\end{aligned}
\tag{21}
$$

$$
\begin{aligned}
\|B_{t+1}\|_F^2 - \|B_t\|_F^2 &= -2\eta\left\langle B_t B_t^\top, \Sigma - A_t A_t^\top + B_t B_t^\top + \frac{K_t^\top K_t + J_t^\top J_t}{2}\right\rangle \\
&\quad -\eta\|A_t B_t^\top - B_t A_t^\top\|_F^2 + \eta\left\langle B_t^\top A_t, K_t^\top K_t - J_t^\top J_t\right\rangle \\
&\quad +\eta^2\|(\Sigma - A_t A_t^\top + B_t B_t^\top)B_t + (A_t B_t^\top - B_t A_t^\top)A_t \\
&\quad -A_t\frac{K_t^\top K_t - J_t^\top J_t}{2} - B_t\frac{K_t^\top K_t + J_t^\top J_t}{2}\|_F^2 \\
&\leq -2\eta\lambda_d(P_t)\|B_t\|_F^2 + \eta\|B_t^\top A_t\|_F\|K_t^\top K_t - J_t^\top J_t\|_F \\
&\quad +\eta^2\|(\Sigma - A_t A_t^\top + B_t B_t^\top)B_t + (A_t B_t^\top - B_t A_t^\top)A_t \\
&\quad -A_t\frac{K_t^\top K_t - J_t^\top J_t}{2} - B_t\frac{K_t^\top K_t + J_t^\top J_t}{2}\|_F^2 \\
&\leq O(\eta e_b^2\varepsilon^2(m+n)d\kappa)\|B_t\|_F^2 + O(\eta\sqrt{\sigma_1}e_b(m+n)d^2\varepsilon^3) \\
&\quad +O(\eta^2\sigma_1^2 e_b^2 d^2\varepsilon^2). \\
&= O(\eta e_b^2\varepsilon^2(m+n)d\kappa)\|B_t\|_F^2 + O(\eta\sqrt{\sigma_1}e_b(m+n)d^2\varepsilon^3).
\end{aligned}
\tag{22}
$$

$$
\begin{aligned}
\Sigma - U_{t+1+T_0}V_{t+1+T_0}^\top &= (I - \eta U_{t+T_0}U_{t+T_0}^\top)(\Sigma - U_{t+T_0}V_{t+T_0}^\top)(I - \eta V_{t+T_0}V_{t+T_0}^\top) \\
&\quad -\eta^2 U_{t+T_0}U_{t+T_0}^\top(\Sigma - U_{t+T_0}V_{t+T_0}^\top)V_{t+T_0}V_{t+T_0}^\top \\
&\quad -\eta^2(\Sigma - U_{t+T_0}V_{t+T_0}^\top)V_{t+T_0}U_{t+T_0}^\top(\Sigma - U_{t+T_0}V_{t+T_0}^\top) \\
&\quad +\eta(U_{t+T_0} + \eta(\Sigma - U_{t+T_0}V_{t+T_0}^\top)V_{t+T_0})J_{t+T_0}^\top J_{t+T_0}V_{t+T_0}^\top \\
&\quad +\eta U_{t+T_0}K_{t+T_0}^\top K_{t+T_0}(V_{t+T_0} + \eta(\Sigma - U_{t+T_0}V_{t+T_0}^\top)^\top U_{t+T_0})^\top \\
&\quad -\eta^2 U_{t+T_0}K_{t+T_0}^\top K_{t+T_0}J_{t+T_0}^\top J_{t+T_0}V_{t+T_0}^\top.
\end{aligned}
\tag{23}
$$

# B    Dynamics in the Symmetric and Full-Rank Case

We consider the case where $U = V = A$ and $\Sigma$ is symmetric and full-rank, and we use gradient flow. We can derive the dynamics of $S = AA^\top$ as $\dot{S} := (\Sigma - S)S + S(\Sigma - S)$, which is a quadratic ordinary differential equation and it is hard to solve directly.

However, if we define $\overline{X} := S^{-1}$, we have $S\overline{X} \equiv I$. Taking the derivative implies $\dot{S}\overline{X} + S\dot{\overline{X}} = 0$. Hence, $\dot{\overline{X}} = -S^{-1}\dot{S}S^{-1}$. Substitute $\dot{S} = (\Sigma - S)S + S(\Sigma - S)$ in it, we have

$$\dot{\overline{X}} = -S^{-1}\left((\Sigma - S)S + S(\Sigma - S)\right)S^{-1} = -\overline{X}\Sigma - \Sigma\overline{X} + 2I,$$

which is a *linear* ordinary differential equation.

For simplicity, define $X := \overline{X} - \Sigma^{-1}$. Then

$$\dot{X} = -X\Sigma - \Sigma X. \tag{24}$$

Solving this equation and we have

$$X(t) = e^{-t\Sigma}X_0 e^{-t\Sigma}. \tag{25}$$

Finally, we could conclude that

$$S(t) = \left(e^{-t\Sigma}(S_0^{-1} - \Sigma^{-1})e^{-t\Sigma} + \Sigma^{-1}\right)^{-1}. \tag{26}$$

Similarly, because $P$'s dynamic is $\dot{P} = -(\Sigma - P)P - P(\Sigma - P)$, we have

$$P(t) = \left(e^{t\Sigma}(P_0^{-1} - \Sigma^{-1})e^{t\Sigma} + \Sigma^{-1}\right)^{-1}, \tag{27}$$

where $P_0 := \Sigma - S_0$.

And it is interesting to verify that $S(t) + P(t) \equiv \Sigma$ by using the following lemma.

**Lemma B.1.** *Suppose $S, P, E \in \mathbb{R}^{d \times d}$ are three positive definite matrices. $\Sigma = S + P$. Suppose $E$ commutes with $\Sigma$. Then*

$$\left(E(S^{-1} - \Sigma^{-1})E + \Sigma^{-1}\right)^{-1} + \left(E^{-1}(P^{-1} - \Sigma^{-1})E^{-1} + \Sigma^{-1}\right)^{-1} = \Sigma.$$

## C  Proof of Lemmas

*Proof of lemma B.1.* Since $\Sigma$ is invertible, we only need to verify the equation after right multiplying both side by $\Sigma^{-1}$. We have

$$\left(E(S^{-1} - \Sigma^{-1})E + \Sigma^{-1}\right)^{-1}\Sigma^{-1} + \left(E^{-1}(P^{-1} - \Sigma^{-1})E^{-1} + \Sigma^{-1}\right)^{-1}\Sigma^{-1}$$

$$= \left(E(\Sigma S^{-1} - I)E + I\right)^{-1} + \left(E^{-1}(\Sigma P^{-1} - I)E^{-1} + I\right)^{-1} \tag{28}$$

$$= \left(E(PS^{-1})E + I\right)^{-1} + \left(E^{-1}(SP^{-1})E^{-1} + I\right)^{-1} \tag{29}$$

$$= (Z + I)^{-1} + (Z^{-1} + I)^{-1} \quad \text{(we denote } E(PS^{-1})E \text{ by } Z \text{ here)} \tag{30}$$

$$= (Z + I)^{-1} + Z(Z + I)^{-1}$$

$$= I$$

$$= \Sigma\Sigma^{-1},$$

where (28) is because $\Sigma$ commutes with $E$, (29) is because $\Sigma = S + P$ and finally (30) is because $\left(E(PS^{-1})E\right)^{-1} = E^{-1}(SP^{-1})E^{-1}$. $\square$

**General analysis for lemma 3.2 and 3.3** Suppose $\bar{S}, \tilde{S}$ and $\Sigma$ are three symmetric matrices. Define $D = \bar{S} - \tilde{S}$. Then we have equation

$$(I + \eta(\Sigma - \bar{S}))\bar{S}(I + \eta(\Sigma - \bar{S})) - (I + \eta(\Sigma - \tilde{S}))\tilde{S}(I + \eta(\Sigma - \tilde{S}))$$

$$= \bar{S} - \tilde{S} + \eta\left((\Sigma - \bar{S})\bar{S} + \bar{S}(\Sigma - \bar{S}) - (\Sigma - \tilde{S})\tilde{S} + \tilde{S}(\Sigma - \tilde{S})\right)$$

$$\quad + \eta^2\left((\Sigma - \bar{S})\bar{S}(\Sigma - \bar{S}) - (\Sigma - \tilde{S})\tilde{S}(\Sigma - \tilde{S})\right)$$

$$= D + \eta((\Sigma - \bar{S} - \tilde{S})D + D(\Sigma - \bar{S} - \tilde{S}))$$

$$\quad + \eta^2\left((\Sigma - \bar{S})\bar{S}(\Sigma - \bar{S}) - (\Sigma - \tilde{S})\tilde{S}(\Sigma - \tilde{S})\right)$$

$$= \left(I + \eta(\Sigma - \bar{S} - \tilde{S})\right)D\left(I + \eta(\Sigma - \bar{S} - \tilde{S})\right)$$

$$\quad + \eta^2\left((\Sigma - \bar{S} - \tilde{S})D(\Sigma - \bar{S} - \tilde{S}) + (\Sigma - \bar{S})\bar{S}(\Sigma - \bar{S}) - (\Sigma - \tilde{S})\tilde{S}(\Sigma - \tilde{S})\right). \tag{31}$$

*Proof of lemma 3.2.* First of all, we can expand the expression of $S'$ and split it in the following terms.

$$\sigma_d(S') \geq \lambda_d \left(\beta S - 2\eta S^2 + \eta^2 S^3\right) + \sigma_d \left((1-\beta)S + \eta\Sigma S + \eta S\Sigma + \frac{\eta^2}{1-\beta}\Sigma S\Sigma\right)$$
$$+\eta^2 \lambda_d \left(-\frac{\beta}{1-\beta}\Sigma S\Sigma - \Sigma SS - SS\Sigma\right).$$

For the first term $\beta S - 2\eta S^2 + \eta^2 S^3$, its eigenvalues are $\beta s_i - 2\eta s_i^2 + \eta^2 s_i^3$ since $S$ is commutable with itself, where $s_i$ is the $i^{\text{th}}$ largest singular value of $S$. By the assumptions $s_i \leq 2\sigma_1$ and $\eta \leq \frac{\beta}{8\sigma_1}$, we see the smallest eigenvalue of $\beta S - 2\eta S^2 + \eta^2 S^3$ is exactly $\beta s - 2\eta s^2 + \eta^2 s^3$.

For the second term, it can be rewritten as

$$(1-\beta)S + \eta\Sigma S + \eta S\Sigma + \frac{\eta^2}{1-\beta}\Sigma S\Sigma \equiv \left(\sqrt{1-\beta}I + \frac{\eta}{\sqrt{1-\beta}}\Sigma\right) S \left(\sqrt{1-\beta}I + \frac{\eta}{\sqrt{1-\beta}}\Sigma\right).$$

Hence, the minimal singular value can be bounded by $\left(\sqrt{1-\beta} + \frac{\eta\sigma_d}{\sqrt{1-\beta}}\right)^2 s$.

Finally, the last term can be lower bounded by $-\eta^2 \sigma_1 \left(-\frac{\beta}{1-\beta}\Sigma S\Sigma - \Sigma SS - SS\Sigma\right) \geq -\frac{8+6\beta}{1-\beta}\eta^2 \sigma_1^3$. Summing up all three terms and we get

$$
\begin{aligned}
s' &\geq (\beta s - 2\eta s^2 + \eta^2 s^3) + \left(\sqrt{1-\beta} + \frac{\eta\sigma_d}{\sqrt{1-\beta}}\right)^2 s - \frac{8+6\beta}{1-\beta}\eta^2\sigma_1^3 \\
&= (1+\eta(\sigma_d - s))^2 s + \frac{\beta\sigma_d^2}{1-\beta}\eta^2 s + 2\sigma_d\eta^2 s^2 - \frac{8+6\beta}{1-\beta}\sigma_1^3\eta^2 \\
&\geq (1+\eta(\sigma_d - s))^2 s - \frac{8+6\beta}{1-\beta}\sigma_1^3\eta^2.
\end{aligned}
$$

$\square$

**Remark:** If we choose $\bar{S} = S$ and $\tilde{S} = \sigma_d(S)I$ in equation (31), we know $D = \bar{S} - \tilde{S} \succeq 0$. Hence $\sigma_d(S') \geq (1+\eta(\sigma_d - s))^2 s - O(\sigma_1^3\eta^2)$.

*Proof of lemma 3.3.* If $p \geq 0$, it suggests that $P$ is positive semi-definite, and $P'$ is positive semi-definite, too. Hence $p' \geq 0$ if $p \geq 0$.

If $p \leq 0$, we can expand the expression of $P'$ and split it in the following terms.

$$\lambda_d(P') \geq \lambda_d\left(\beta P + 2\eta P^2 + \eta^2 P^3\right) + \lambda_d\left((1-\beta)P - \eta\Sigma P - \eta P\Sigma + \frac{\eta^2}{1-\beta}\Sigma P\Sigma\right)$$
$$+\eta^2 \lambda_d\left(-\frac{\beta}{1-\beta}\Sigma P\Sigma - \Sigma PP - PP\Sigma\right).$$

For the first term $\beta P + 2\eta P^2 + \eta^2 P^3$, its eigenvalues are $\beta p_i + 2\eta p_i^2 + \eta^2 p_i^3$ since $P$ is commutable with itself, where $p_i$ is the $i^{\text{th}}$ largest eigenvalue of $P$. By the assumptions $|p_i| \leq 2\sigma_1$ and $\eta \leq \frac{\beta}{8\sigma_1}$, we see the smallest eigenvalue of $\beta P + 2\eta P^2 + \eta^2 P^3$ is exactly $\beta p + 2\eta p^2 + \eta^2 p^3$.

For the second term, it can be rewritten as

$$(1-\beta)P - \eta\Sigma P - \eta P\Sigma + \frac{\eta^2}{1-\beta}\Sigma P\Sigma \equiv \left(\sqrt{1-\beta}I - \frac{\eta}{\sqrt{1-\beta}}\Sigma\right) P \left(\sqrt{1-\beta}I - \frac{\eta}{\sqrt{1-\beta}}\Sigma\right).$$

Hence, the minimal eigenvalue can be bounded by $\left(\sqrt{1-\beta} - \frac{\eta\sigma_d}{\sqrt{1-\beta}}\right)^2 p$ if $p \leq 0$.

Finally, the last term can be lower bounded by $-\eta^2\sigma_1\left(-\frac{\beta}{1-\beta}\Sigma P\Sigma - \Sigma PP - PP\Sigma\right) \geq -\frac{8+6\beta}{1-\beta}\eta^2\sigma_1^3$. Summing up all three terms and we get that when $p \leq 0$,

$$
\begin{aligned}
p' &\geq (\beta p + 2\eta p^2 + \eta^2 p^3) + \left(\sqrt{1-\beta} - \frac{\eta\sigma_d}{\sqrt{1-\beta}}\right)^2 p - \frac{8+6\beta}{1-\beta}\eta^2\sigma_1^3 \\
&= (1-\eta(\sigma_d - p))^2 p + \frac{\beta\sigma_d^2}{1-\beta}\eta^2 p + 2\sigma_d\eta^2 p^2 - \frac{8+6\beta}{1-\beta}\sigma_1^3\eta^2 \\
&\geq (1-\eta(\sigma_d - p))^2 p - \frac{8+6\beta}{1-\beta}\sigma_1^3\eta^2 \\
&\geq (1-\eta\sigma_d)^2 p - \frac{8+6\beta}{1-\beta}\sigma_1^3\eta^2.
\end{aligned}
$$

$\square$

**Remark:** Similarly, if we choose $\bar{S} = P$ and $\tilde{S} = \lambda_d(P)I$ in equation (31), we have $D = P - \lambda_d(P)I \succeq 0$. Hence we have $\lambda_d(P') \geq \min\left\{0, (1-\eta\sigma_d)^2 p + O(\sigma_1^3\eta^2)\right\}$.

## D  Solving the Iteration Formula of $a$

In this section we analyze the iteration formula (15).

We first consider the case when $a_t \leq \sqrt{\frac{\sigma_d}{2}}$. Notice that $a_t \geq \frac{\varepsilon}{c\sqrt{d}}$, we have

$$
\sqrt{(1+\eta(\sigma_d - \sigma_d(A_t)^2))^2\sigma_d(A_t)^2 - 22\sigma_1^3\eta^2} \geq (1+\eta(\sigma_d - \sigma_d(A_t)^2))\sigma_d(A_t) - 22\frac{c\sqrt{d}}{\varepsilon}\sigma_1^3\eta^2,
$$

where we choose $\eta$ so small that $22\sigma_1^3\eta^2 \leq \frac{\varepsilon^2}{c^2 d}$.

By taking $\varepsilon = O\left(\frac{\sigma_d}{\sqrt{d^3\sigma_1 e_b^2(m+n)}}\right)$ and $\eta = O\left(\frac{\sigma_d\varepsilon^2}{d\sigma_1^3}\right)$, we have $\frac{1}{2}\eta(\sigma_d - a_t^2)a_t \geq \frac{1}{2}\eta\frac{\sigma_d}{2}\frac{\varepsilon}{c\sqrt{d}} \geq 22\frac{c\sqrt{d}}{\varepsilon}\sigma_1^3\eta^2 + 1.5\sqrt{2\sigma_1}\eta(e_b^2 + c^2)\varepsilon^2(m+n)d$, hence,

$$
a_{t+1} \geq \left(1 + \frac{\eta}{2}(\sigma_d - a_t^2)\right)a_t, \tag{32}
$$

and

$$
s_{t+1} \geq \left(1 + \frac{\eta}{2}(\sigma_d - s_t)\right)^2 s_t \geq (1 + \eta(\sigma_d - s_t))s_t. \tag{33}
$$

Subtracting $\sigma_d$ by (33), we have

$$
\sigma_d - s_{t+1} \leq (1 - \eta s_t)(\sigma_d - s_t). \tag{34}
$$

Dividing (33) by (34) we have

$$
\frac{s_{t+1}}{\sigma_d - s_{t+1}} \geq \frac{1 + \eta(\sigma_d - s_t)}{1 - \eta s_t}\frac{s_t}{\sigma_d - s_t} \geq (1 + \eta\sigma_d)\frac{s_t}{\sigma_d - s_t}.
$$

Hence, $\frac{s_T}{\sigma_d - s_T} \geq (1 + \eta\sigma_d)^T\frac{s_0}{\sigma_d}$. So, it takes at most $T_1 := O\left(\frac{1}{\eta\sigma_d}\ln\frac{d\sigma_d}{\varepsilon^2}\right)$ iterations to bring $a_t$ to at least $\sqrt{\frac{\sigma_d}{2}}$.

## E  Solving the Iteration Formula on $B$

The iteration formula can be summarized as

$$
\|B_{t+1}\|_F^2 \leq (1 + p)\|B_t\|_F^2 + q,
$$

where $p = O(\eta e_b^2 \varepsilon^2 (m+n)d\kappa)$ and $q = O(\eta\sqrt{\sigma_1}e_b(m+n)d^2\varepsilon^3)$. Moreover, we have

$$\|B_T\|_F^2 \le (1+p)^T\|B_0\|_F^2 + ((1+p)^T - 1)\frac{q}{p}.$$

Suppose $T \le T_0 = O\left(\frac{1}{\eta\sigma_d}\ln\frac{d\sigma_d}{\varepsilon^2}\right)$. By choosing $\varepsilon = \tilde{O}\left(\frac{\sqrt{\sigma_d}}{e_b\sqrt{(m+n)d\kappa}}\right)$ [12], we have $pT \le pT_0 \le$ 1. Then $(1+p)^T = 1+\binom{T}{1}p+\binom{T}{2}p^2+\cdots+\binom{T}{T}p^T \le 1+Tp\left(1+\frac{1}{2!}+\cdots+\frac{1}{T!}\right) \le 1+(e-1)Tp \le 1+2pT$. Hence,

$$\|B_T\|_F^2 \le (1+2pT)\|B_0\|_F^2 + 2qT.$$

Similarly, by choosing $\varepsilon = \tilde{O}\left(\frac{\sigma_d}{\sqrt{\sigma_1}e_b(m+n)}\right)$, we have $qT \le c^2d^2\varepsilon^2$. By taking $e_b = 2c$, we have

$$\|B_T\|_F^2 \le 3\|B_0\|_F^2 + c^2d^2\varepsilon^2 \le 4c^2d^2\varepsilon^2 = e_b^2d^2\varepsilon^2,$$

induction succeeds.

## F  Proof of Stage Two

Here is the full version of the proof. Initially, $\|\Delta_0\|_{op} = \|P_{T_0} + Q_{T_0}\|_{op}$ where $Q = AB^\top - BA^\top$. Hence $\|\Delta_0\|_{op} \le \sigma_1(P_{T_0}) + \sigma_1(Q_{T_0}) \le \frac{\sigma_d}{4} + \sqrt{2\sigma_1}\sigma_1(B_{T_0}) \le \frac{\sigma_d}{3}$. Then for $U_{T_0}$ we have $\frac{2\sigma_d}{3} \le \sigma_d(\Sigma) - \sigma_1(\Delta_0) \le \sigma_d(U_{T_0}V_{T_0}^\top) \le \sigma_d(U_{T_0}U_{T_0}^\top) - 2\sigma_1(U_{T_0}B_{T_0}^\top) \le \sigma_d(U_{T_0}U_{T_0}^\top) - 4\sqrt{2\sigma_1}O\left(\frac{\sigma_d}{\sqrt{\sigma_1}}\right)$. Hence $\sigma_d(U_{T_0}) \ge \sqrt{\frac{\sigma_d}{2}}$. We can do the same thing on $V_{T_0}$.

First of all, by equations (8) and (9), we have

$$\|J_{t+T_0}\|_{op} \le c\varepsilon\left(1 - \frac{\eta\sigma_d}{2}\right)^t\sqrt{\max\{m', d\}},$$

and

$$\|K_{t+T_0}\|_{op} \le c\varepsilon\left(1 - \frac{\eta\sigma_d}{2}\right)^t\sqrt{\max\{n', d\}}.$$

Expanding $\Sigma - U_{t+1+T_0}V_{t+1+T_0}^\top$ by brute force [13], we get

$$
\begin{aligned}
\Delta_{t+1} &\le \left(1 - \frac{\eta\sigma_d}{2}\right)^2\Delta_t + O(\eta^2\sigma_1^2)\Delta_t + O\left(\eta\varepsilon^2\sigma_1(m+n)\right)\left(1 - \frac{\eta\sigma_d}{2}\right)^{2t} \\
&\le \left(1 - \frac{\eta\sigma_d}{2}\right)\Delta_t + O\left(\eta\varepsilon^2\sigma_1(m+n)\right)\left(1 - \frac{\eta\sigma_d}{2}\right)^{2t}.
\end{aligned}
$$

Then,

$$
\begin{aligned}
\frac{\Delta_{t+1}}{\left(1 - \frac{\eta\sigma_d}{2}\right)^{t+1}} &\le \frac{\Delta_t}{\left(1 - \frac{\eta\sigma_d}{2}\right)^t} + O\left(\eta\varepsilon^2\sigma_1(m+n)\right)\left(1 - \frac{\eta\sigma_d}{2}\right)^{t-1} \\
&\le \Delta_0 + O\left(\varepsilon^2\kappa(m+n)\right) \\
&\le \frac{2}{5}\sigma_d.
\end{aligned}
$$

Thus we can now verify that $\Delta_t \le \left(1 - \frac{\eta\sigma_d}{2}\right)^t\frac{2}{5}\sigma_d$. Together with the linear convergence of $J$ and $K$, we know the gradient descent converge linearly. Notice that by using the operator norm of $\Delta_t$, we can easily prove that $\sigma_d(U)$ and $\sigma_d(V)$ in the next iteration is at least $\sqrt{\frac{\sigma_d}{2}}$ once given $\|B_{T_0+t}\|_F$ is small.

To give an upper bound on $\|B\|_F$, we still use equation (19).

First of all, we have $\|P\|_F^2 + \|Q\|_F^2 = \|\Sigma - UV^\top\|_F^2$, since $P + Q = \Sigma - UV^\top$, and $\langle P, Q\rangle = 0$. Hence, $\|P_{t+T_0}\|_F \le \sqrt{d}\Delta_t$ and $\|Q_{t+T_0}\|_F \le \sqrt{d}\Delta_t$.

---

[12]Here $\tilde{O}$ means there might be some log terms about $m, n, \kappa$ and $e_b$ on the denominator.

[13]Please see (23) for the result of the expanding.

Finally,

$$
\begin{aligned}
\|B_{t+1+T_0}\|_F^2 &\leq \left(1 + 2\eta\left(1 - \frac{\eta\sigma_d}{2}\right)^t \frac{2}{5}\sigma_d\right)\|B_{t+T_0}\|_F^2 + O\left(\eta\sigma_d(m+n)d\varepsilon^2\right)\left(1 - \frac{\eta\sigma_d}{2}\right)^{2t} \\
&\quad + O\left(\eta^2 d\sigma_1\sigma_d^2\left(1 - \frac{\eta\sigma_d}{2}\right)^{2t}\right).
\end{aligned}
$$

To solve this iteration formula, we first notice that the product of the main coefficient is bounded by a universal constant,

$$
\Xi_T := \prod_{i=0}^{T-1}\left(1 + 2\eta\left(1 - \frac{\eta\sigma_d}{2}\right)^t \frac{2}{5}\sigma_d\right) \leq \exp\left(\sum_{i=0}^{T-1} 2\eta\left(1 - \frac{\eta\sigma_d}{2}\right)^t \frac{2}{5}\sigma_d\right) \leq e^{\frac{8}{5}},
$$

we can then write it into an iteration formula about $\frac{\|B_{t+T_0}\|_F^2}{\Xi_t}$,

$$
\begin{aligned}
\frac{\|B_{t+1+T_0}\|_F^2}{\Xi_{t+1}} &\leq \frac{\|B_{t+T_0}\|_F^2}{\Xi_t} + O\left(\eta\sigma_d(m+n)d\varepsilon^2\right)\left(1 - \frac{\eta\sigma_d}{2}\right)^{2t} \\
&\quad + O\left(\eta^2 d\sigma_1\sigma_d^2\left(1 - \frac{\eta\sigma_d}{2}\right)^{2t}\right) \\
&\leq \|B_{T_0}\|_F^2 + O\left((m+n)d\varepsilon^2\right) + O\left(\eta d\sigma_1\sigma_d\right).
\end{aligned}
$$

By taking $\varepsilon = O\left(\frac{\sigma_d}{\sqrt{\sigma_1(m+n)d}}\right)$ and $\eta = O\left(\frac{\sigma_d}{d\sigma_1^2}\right)$, induction on $\|B\|_F$ holds.

# G  Matrix sensing problem

We only consider full-rank case here, i.e. $\Sigma$ is a $d \times d$ full-rank matrix, and we would like to factorize $\Sigma$ into $U \times V^\top$, where $U, V \in \mathbb{R}^{d \times d}$.

For a sufficiently large integer $N$, consider measurements $M_1, M_2, \cdots, M_N \in \mathbb{R}^{d \times d}$ generated by i.i.d. Gaussian distribution. Define labels $y_i := \langle M_i, \Sigma\rangle$ for $i \in [N]$.

The objective function is defined as

$$
f(U, V) = \frac{1}{2N}\sum_{i \in [N]}\left(\langle M_i, UV^\top\rangle - y_i\right)^2,
$$

which can be equivalently written as $\frac{1}{2N}\sum_{i \in [N]}\langle M_i, UV^\top - \Sigma\rangle^2$.

And the gradient descent with learning rate $\eta$ can be written as

$$
\begin{aligned}
U_{t+1} &= U_t - \frac{\eta}{N}\sum_{i \in [N]}\langle M_i, UV^\top - \Sigma\rangle M_i V; \\
V_{t+1} &= V_t - \frac{\eta}{N}\sum_{i \in [N]}\langle M_i, UV^\top - \Sigma\rangle M_i^\top U.
\end{aligned}
$$

## G.1  Symmetrization

Suppose the SVD of $\Sigma$ is $\Phi\Sigma'\Psi^\top$. Then if we replace the objective matrix by $\Sigma'$, replace the measurements by $\Phi^\top M_i\Psi$ and replace the initial parameter matrices by $\Phi^\top U$ and $\Psi^\top V$, then everything, including the objective function, the gradient descent process, the loss value, etc. are the same. Hence, we can assume, without loss of generality, $\Sigma$ is a positive semi-definite matrix. (We could also check that the initialization and measurements are still i.i.d. Gaussian generated.)

To simplify the notation, we define a linear operator $\Lambda : \mathbb{R}^{d \times d} \to \mathbb{R}^{d \times d}, \Lambda(X) := \frac{1}{N}\sum_{i \in [N]}\langle M_i, X\rangle M_i$. A standard concentration analysis shows that when there are sufficiently large number of measurements, then with large probability, $\Lambda$ is sufficiently close to an identity operator,

with respect to operator norm. Define the error term $E : \mathbb{R}^{d \times d} \to \mathbb{R}^{d \times d}, E(X) = \Lambda(X) - X$. The error term $E$ can be described by RIP.

Hence, the gradient process can now be written as

$$
\begin{aligned}
U_{t+1} &= U_t - \eta(U_t V_t^\top - \Sigma)V_t - \eta E(U_t V_t^\top - \Sigma)V_t; \\
V_{t+1} &= V_t - \eta(U_t V_t^\top - \Sigma)^\top U_t - \eta E(U_t V_t^\top - \Sigma)^\top U_t.
\end{aligned}
$$

Hence, we can define $A_t = \frac{U_t + V_t}{2}$ and $B_t = \frac{U_t - V_t}{2}$. Then the iteration formula becomes

$$
\begin{aligned}
A_{t+1} &= A_t + \eta(\Sigma - A_t A_t^\top + B_t B_t^\top - E_t^+)A_t - \eta(A_t B_t^\top - B_t A_t^\top - E_t^-)B_t; \\
B_{t+1} &= B_t - \eta(\Sigma - A_t A_t^\top + B_t B_t^\top + E_t^-)B_t + \eta(A_t B_t^\top - B_t A_t^\top + E_t^+)A_t,
\end{aligned}
$$

where $E_t^+ = \frac{E(U_t V_t^\top - \Sigma) + E(U_t V_t^\top - \Sigma)^\top}{2}$ and $E_t^- = \frac{E(U_t V_t^\top - \Sigma) - E(U_t V_t^\top - \Sigma)^\top}{2}$ are small matrices.

By lemma 3.2 we know that if $B$ and $E^{+/-}$ is small, the minimal singular value of $A$ is monotonically increasing. And similarly, we could define $P$ and hopefully we could also use lemma 3.3 to prove that the minimal eigenvalue of $P$ is not very small and hence the F-norm of $B$ won't be too large.

**Remark G.2.** *As for deep matrix factorization problem, there could be some similar techniques to handle it. For instance, if we would like to factorize $\Sigma$ into $2m$ matrices $\prod_{i \in [2m]} U_i$, one naive idea is to first symmetrize $\Sigma$ and then define $A_i = \frac{U_i + U_{2m-i}}{2}$ and $B_i = \frac{U_i - U_{2m-i}}{2}$ for $i \in [m]$. If we can find any monotonic value in these matrices (possibly the minimal singular values of $A_i$), it would guide us to the global convergence.*