# OpenReview forum: "Global Convergence of Gradient Descent for Asymmetric Low-Rank Matrix Factorization"
_NeurIPS.cc/2021/Conference — NeurIPS 2021 Poster_

### Official Review · Reviewer_UcqW · 2021-07-05

**Rating:** 9
**Confidence:** 4

**Summary:**

This paper provides a proof of polynomial convergence rate whp of randomly initialized gradient descent on the non-convex problem of matrix factorization with the Frobenius norm. Notably, this problem was considered in previous works and only weaker results have been shown, e.g. no convergence rate, or polynomial convergence rate with the addition of noise and regularization.

**Limitations And Societal Impact:**

Yes.

**Main Review:**

The related work section provides appropriate context for understanding the problem and the paper's contribution.
Overall, the paper is very well structured for guiding the reader through the verification of the proof. The authors did a great job with how cleanly and clearly they broke down such a technically heavy proof into individual parts. I was able to follow most of the proof, and did not find any errors. (I assume in line 459, in the supplementary material, the \sqrt{d^3} in the denominator is meant to be \sqrt{d}, as this matches the original definition of epsilon and still works in the following lines)

There are some minor typos/grammatical errors but they do not detract from understanding.

In terms of the proof itself, I do not feel like I have a good intuitive understanding of what the proof is doing, more specifically what exactly is the main idea behind it, if there is one. I don't mean to say it is technically lacking, rather it is much the opposite as I alluded to above; there is a lot of careful balancing of terms and algebraic manipulation that leverages the specific choices of the noise variance and learning rate in multiple ways, which seems extremely nontrivial to come up with. Suffice it to say, I am not surprised why there were several works on this problem before and apparently none of them were able to solve it without weakening the result in some way. I was not able to see how the PL inequality the authors mentioned is relevant to this specific problem, though maybe that is due to a lack of experience in optimization literature on my part. It just seems like the authors wait for certain quantities to reach certain values, and then once that point is reached a decay factor can be upper bounded and a bunch of perturbation terms can be absorbed/canceled. Again, finding the right terms to pay attention to like the authors did seems extremely nontrivial.  On a more high level aspect of the proof, I found the reduction to the diagonal case and the splitting of the U,V matrices into U/J, V/K very nice.

Perhaps the intuition of the proof machinery could be better conveyed, but because of the significance and difficulty of the problem the authors solved, as of now I rate this paper between an 8 and a 9. To me it clearly deserves to be published at a NeuRIPs-tier conference, but the problem it solves maybe seems like a bit of a toy problem, despite the clear interest in it by the theoretical community demonstrated by previous work. It was not clear whether the result of the paper has any bearing on or could have its proof be extended to something outside of this specific matrix factorization problem.

**Time Spent Reviewing:**

10

---

> ### Author Response · Authors · 2021-08-10
> **Response to Reviewer UcqW**
>
> We thank the reviewer for the valuable feedback.
>
> **Typos**: We will correct typos in our final version.
>
>
> **Main proof ideas**:
> The main proof ideas are presented in Section 2. Indeed, some proof techniques are difficult to explain without formulas. For PL-inequality, it is known that near the global minimum, if $U$ and $V$ are balanced, then we have this condition. See e.g.,  our Stage 2 and prior work (Ma et al. 2021). We will extend Section 2 to add more explanations about PL-inequality and our main proof ideas in the final version.
>
>
> **Generalization beyond matrix factorization**: We choose to focus on asymmetric matrix factorization because it is the most canonical setting that admits the phenomenon we would like to understand. We believe our techniques can be extended to other problems. For example, for matrix sensing, if the sampling matrices satisfy RIP, we believe the monotonicity of $\sigma_d(A)$ and the slow growth of $\|\|B\|\|_F$ still hold and can be analyzed using our techniques.

---

### Official Review · Reviewer_RZoC · 2021-07-07

**Rating:** 7
**Confidence:** 4

**Summary:**

This paper studies the performance of the gradient descent (GD) algorithm for a prototypical instance of the asymmetric matrix factorization (AMF) problem. The main result of the paper is establishing the polynomial convergence of GD for an ideal and noiseless, but unregularized instance of the asymmetric matrix factorization. To the best of my knowledge, the technical contributions of the paper are sound and correct.

**Limitations And Societal Impact:**

The authors are encouraged to address the above-mentioned comments.

**Main Review:**

While I believe that the technical contribution of the paper is novel and clear, I do have several comments:

1. My major comment is that the proposed analysis of this work seem to only work for an idealized objective function. Indeed, the practical instances of the MF problems mostly rely on linear measurements of the form $y_i = trace(A_i’ X)$. Indeed, the considered loss function in this paper corresponds to the ideal population loss function, where the number of available samples approach infinity (or equivalently, the so-called RIP condition holds with constant zero). This would not pose any issue if the authors had clearly explained how their proposed technique can be extended beyond this synthetic variant of the problem. Based on my understanding, and unlike the authors’ claim, I don’t think their provided technique can be directly used in more general instances of AMF because these problems are not diagonalizable via a change of basis (i.e., Eq. 4 and 5) when the measurements are of the form $y_i = trace(A_i’ X)$. The authors should support their claim by explaining how their proposed technique can be extended to other practical instances of AMF, as well as “other related non-convex problems” (as claimed in the abstract).

2. While I think relaxing the requirement for regularization is one of the main contributions of this work, I’d like to point out that another reason for such regularization is to remove optimal solutions at infinity. Indeed, without a regularizer that balances the norm of the components, globally optimal solutions exist at infinity. On the other hand, Theorem 1.1 and Corollary 1.2 only guarantee the convergence of the objective function, and do not provide any bound on the values of the obtained solution. Is it possible to show that $U_t$ and $V_t$ do not diverge if we run the algorithm for a longer time?

3. Finally, it appears to me that the paper is not polished, and it suffers from numerous grammatical and notational mistakes. While these mistakes are mostly minor and can be fixed, they severely undermine the readability of the paper. For instance, some notations are used before their formal definitions (e.g. A and B in page 4).


===================================================
After rebuttal, the reviewers have successfully addressed my comments. I strongly encourage the authors to include their suggested method for handling the general linear measurements to the revised paper.


**Time Spent Reviewing:**

4

---

> ### Author Response · Authors · 2021-08-10
> **Response to Reviewer RZoC**
>
>
> We thank the reviewer for appreciating our technical contributions. Please find our responses to your comments below:
>
> **Extension to matrix sensing**:
> We consider the matrix sensing problem with the objective function: $
> f(U, V) = \frac{1}{2N}\sum\limits_{i\in[N]}(\left\langle M_i, U V^\top\right\rangle - y_i)^2
> $ where $y_i:=\left\langle M_i, \Sigma\right\rangle$, $i\in[N]$, $M_i$s are sensing matrices and $\Sigma$ is the underlying low-rank matrix.
> We can use the same arguments in line 125 - 127: Suppose the SVD of $\Sigma$ is $\Phi\Sigma'\Psi^\top$. Let $U_t’ = \Phi^{-1}U_t $ and $V_t’ = \Psi^{-1} V_t$. We also transform the sensing matrix $M_i’ = \Phi^\top M_i\Psi$. Now we can work with $\Sigma’, U_t’, V_t’, M_i’$  and  obtain analogue dynamics of Eq. 4 and Eq. 5:
> $
>     U_{t+1}’=U_t’-\eta(U_t’V_t’^\top - \Sigma’)V_t’ - \eta \mathcal{E}(U_t’V_t’^\top - \Sigma’)V_t’;\notag\\
>     V_{t+1}’=V_t’-\eta(U_t’V_t’^\top - \Sigma’)^\top U_t’ - \eta \mathcal{E}(U_t’V_t’^\top - \Sigma’)^\top U_t’
> $
> where $\mathcal{E}(\cdot)$ is an operator: $\mathcal{E}(X): \frac{1}{N}\sum\limits_{i\in[N]}\left\langle M_i’, X\right\rangle M_i’ - X$.
> Note we only have two additional terms that related to the operator $\mathcal{E}(\cdot)$: $ \eta \mathcal{E}(U_t’V_t’^\top - \Sigma’)V_t’$ and $ \eta \mathcal{E}(U_t’V_t’^\top - \Sigma’)^\top U_t’$. Therefore, we can use the same techniques in our paper to analyze the dynamics $U_t’-\eta(U_t’V_t’^\top - \Sigma)V_t’$ and $V_t’-\eta(U_t’V_t’^\top - \Sigma)^\top U_t$. The two terms related to $\mathcal{E}(\cdot)$ are perturbation terms due to sampling and the common way for analysis is to relate them to RIP. We leave it as a future work to study the impact of sampling in this dynamics.
>
>
>
>
> **$U_t$ and $V_t$ will not diverge**:
> We thank the reviewer for raising this question. Our analysis implies the convergence of $U_t$ and $V_t$ to finite values as $t \rightarrow \infty$ and thus it is not necessary to add the balancing regularizer. The proof is straightforward:
>
> First of all, because $\|\|B\|\|_F$ is bounded (the first condition in section 3.3) and loss is bounded (the second condition in section 3.3), we obtain upper bounds on $\|\|A\|\|_F$ by $\|\|AA^\top\|\|_F\leq\|\|\Sigma - AA^\top + BB^\top\|\|_F + \|\|\Sigma + BB^\top\|\|_F \leq \|\|\Sigma - UV^\top\|\|_F + \|\|\Sigma\|\|_F + \|\|BB^\top\|\|_F$. Hence we also obtain upper bounds on both $\|\|U\|\|_F$ and $\|\|V\|\|_F$ by the triangle inequality.
>
> On top of that, by equations (6) and (7) in our paper, we know that $\|\|U_{t+1}-U_t\|\|_F$ decreases to zero linearly, since both $\|\|\Sigma - UV^\top\|\|_F$ and $\|\|K\|\|_F$ are exponentially shrinking, and $\|\|U\|\|_F$ and $\|\|V\|\|_F$ are bounded. The same statement holds for the $V_t$ sequence. Finally, we can conclude that the gradient descent trajectory forms a Cauchy sequence, and convergence on $U, V, J$ and $K$ follows.
>
> We will add a formal statement in the final version.
>
>
> **Grammatical and notational mistakes**: We will fix these mistakes.

---

> > ### Comment · Reviewer_RZoC · 2021-08-20
> > **Update on my comments**
> >
> > I appreciate the authors' effort in successfully addressing my comments. I will raise my rating accordingly.

---

### Official Review · Reviewer_b8gy · 2021-07-08

**Rating:** 8
**Confidence:** 3

**Summary:**

The paper shows the global convergence of the matrix factorization problem with geometric convergence speed.

**Limitations And Societal Impact:**

The paper confirms the well-known observation with theoretical proof that the standard gradient descent algorithm finds the global optimum for the matrix factorization problem. I do not think there is any negative societal impact of this work.

**Main Review:**

I think it is a well-written paper. In prior work, while we know the landscape of matrix factorization problem is nice, all proofs (that I know of) for showing global convergences require some artificial modifications on the algorithm, e.g. additional regularizer. It is nice to have proof that does not need those artificial changes. Furthermore, the paper also reveals that while the eigenvalue of U and V may not behave well, the eigenvalues of A=(U+V)/2 and B=(U-V)/2 do behave well. Hence, this paper not only gives a cleaner proof but also adds some insights to the matrix factorization problem.
Looking at the submission history, I think proving geometric convergence is important to the problem (otherwise it still does not fully explain the good performance of the standard grandient descent in practice) and I am glad that the authors have proved it in this new submission.

**Time Spent Reviewing:**

6

---

> ### Author Response · Authors · 2021-08-10
> **Response to Reviewer b8gy**
>
> We thank the reviewer for the positive review!

---

### Official Review · Reviewer_tM8e · 2021-07-14

**Rating:** 6
**Confidence:** 2

**Summary:**

This paper studies the problem of decomposing an m x n matrix of rank r into the product of an m x r matrix and an r x n matrix. While various methods exits for solving this problem, the goal of the paper is to provide theoretical guarantees for gradient descent with random initialization and fixed learning rate. The main result gives sufficient conditions, in terms of the dimensions (m,n), the rank r, and the condition number of the underlying matrix, under which the gradient descent method converges to a global optimum with a polynomial rate.

This work improves upon prior work which either: focused only the symmetric setting, does not give convergence results, or considered different initializations. The analysis and the conditions are based on two different stages. In the first stages,  the iterates converge slowly to get to a point `````near to an optimal solution. In the second stage, the iterate convergence quickly.

No numerical results are provided.

**Limitations And Societal Impact:**

The assumptions for the theoretical results are clearly stated. Given the theoretical nature of this work I do not see the need for any more discussion on societal impact.

**Main Review:**

The authors do a good job of describing their problem and the contribution of the results in the view of prior work. The main results is clearly explained and the high level idea behind the proof a conveyed. The more technical parts are a bit more difficult for me to follow, in part because the transition from continuous time to discrete was not clear to me, and also the convention of removing the time indices from the expressions made things more difficult for me to parse.

I agree with the authors that the main contribution of this paper is the analysis of the method. Indeed, other efficient solutions exist and similar results have been proven for the symmetric setting. Unfortunately, it is difficult for me to describe that precisely the new ideas in the analysis are. This is partly because I am not super familiar with the immediate prior work I do not know which of the steps are novel and which are standard. The authors could do a better job of making this distinction for readers such as myself.

While there is certainly a great deal of work in the literature focusing on variations of this low-rank factorization problem (both in the symmetric and non symmetric settings) it seems to me that it would be much more practically interesting to consider the setting where the matrix is only approximately low rank. I wonder to what extent the insights and analysis from the current line of work extend to this setting.

 more comments:

- The distinction between sigma(.) as a function (applied to a matrix) as opposed to the singular values of a specific matrix (namely Sigma) was confusing, especially with respect to the statement of Lemma 3.2.

- The wording in footnote 5 could be made more precise.

- The matrices and A and B and sort-of-but-not-really defined before they are used, and then they are defined later.






**Time Spent Reviewing:**

3

---

> ### Author Response · Authors · 2021-08-10
> **Response to Reviewer tM8e**
>
> We thank you for the constructive review. Please find our responses to your comments below.
>
>
>
> **Technical Novelties**:
> Our new ideas in the analyses are discussed in Section 2. In particular, the symmetrization technique is new in the literature. For the proof, we also obtain bounds for approximating matrix derivatives (Lemma 3.2 and Lemma 3.3) which are also new. Furthermore, we introduce a new important quantity $P:=\Sigma - AA^\top + BB^\top$ which enjoys a good property: $- \min (0, \lambda_d(P) )$ is small. The matrix $P$ serves not only as a metric between $\Sigma$ and $UV^\top$, but also an upper bound of the growth rate of $\|\|B\|\|_F$. We will emphasize more about our technical novelties in the final version.
>
>
> **Comparison to the Symmetric Case**:
> The symmetric factorization problem (i.e., the case when $U=V$) is much easier since the minimal singular value of $U$ is monotonically increasing and the dynamic of $UU^\top$ can be written in a nice form (see the last equation in page 13).
>
> **Approximately Low Rank**: We believe our techniques still apply to the setting where $\Sigma$ is approximately low-rank and the gap between $\sigma_d$ and $\sigma_{d+1}$ is above a certain threshold. This condition ensures the operator norm of $J$ and $K$ are sufficiently smaller than the minimum singular value of $A$, and our analyses will still hold.
>
> **Writing**: Thanks for your reminder. For the continuous-time case, $U=U(t)$ and $V=V(t)$ are two matrix functions with $t\in\mathbb{R}$ and  $\dot{U}$ means $\frac{\partial U}{\partial t}$. We will add this definition in the introduction section. We will also address the comments on sigma(.), footnote 5, and matrices A and B. Thanks for mentioning these.

---

> > ### Comment · Reviewer_tM8e · 2021-09-12
> > **feedback**
> >
> > Thanks for responding to my comments. I am updated my score.

---

### Decision · Program_Chairs · 2021-09-27

**Decision:**

Accept (Poster)

**Comment:**

We thank the authors for this submission. Overall, the paper presents the first proof that shows randomly initialized gradient descent converges to a global minimum of the asymmetric low-rank factorization problem with a polynomial rate.

The paper well-motivates the approach. The authors have provided extensive responses to the concerns raised (matrix sensing connection, convergence of Ut, Vt, better presentation of results, etc) and the AC + reviewers really thank them for their effort. Overall, the new results obtained during the rebuttal definitely improve the quality of the paper. We all believe that the inclusion of these results during the rebuttal period is something that does not heavily change the message of this paper.

There was discussion and consensus that this work is interesting. Having in mind issues/concerns raised by the reviewers, the main points of reviewers during further discussion were that this paper deserves publication, given the promised fixes by the authors during the discussion period.